# Tumor Biology and Microenvironment of Vestibular Schwannoma-Relation to Tumor Growth and Hearing Loss

**DOI:** 10.3390/biomedicines11010032

**Published:** 2022-12-23

**Authors:** Michaela Tesařová, Lenka Peterková, Monika Šťastná, Michal Kolář, Lukáš Lacina, Karel Smetana, Radovan Hynek, Jan Betka, Aleš Vlasák, Petr Lukeš, Zdeněk Fík

**Affiliations:** 1Department of Otorhinolaryngology and Head and Neck Surgery, 1st Faculty of Medicine, Charles University and Motol University Hospital, 150 06 Prague, Czech Republic; 2Institute of Anatomy, 1st Faculty of Medicine, Charles University, 128 00 Prague, Czech Republic; 3Institute of Molecular Genetics, Czech Academy of Sciences, 142 20 Prague, Czech Republic; 4BIOCEV, Biotechnology and Biomedicine Centre, 252 50 Vestec, Czech Republic; 5Department of Dermatovenereology, 1st Faculty of Medicine, Charles University and General University Hospital, 128 08 Prague, Czech Republic; 6Institute of Biochemistry and Microbiology, University of Chemistry and Technology, 166 28 Prague, Czech Republic; 7Department of Neurosurgery, 2nd Faculty of Medicine, Charles University, 150 06 Prague, Czech Republic

**Keywords:** VS, tumor microenvironment, tumor growth, hearing loss

## Abstract

Vestibular schwannoma is the most common benign neoplasm of the cerebellopontine angle. It arises from Schwann cells of the vestibular nerve. The first symptoms of vestibular schwannoma include hearing loss, tinnitus, and vestibular symptoms. In the event of further growth, cerebellar and brainstem symptoms, along with palsy of the adjacent cranial nerves, may be present. Although hearing impairment is present in 95% of patients diagnosed with vestibular schwannoma, most tumors do not progress in size or have low growth rates. However, the clinical picture has unpredictable dynamics, and there are currently no reliable predictors of the tumor’s behavior. The etiology of the hearing loss in patients with vestibular schwannoma is unclear. Given the presence of hearing loss in patients with non-growing tumors, a purely mechanistic approach is insufficient. A possible explanation for this may be that the function of the auditory system may be affected by the paracrine activity of the tumor. Moreover, initiation of the development and growth progression of vestibular schwannomas is not yet clearly understood. Biallelic loss of the *NF2* gene does not explain the occurrence in all patients; therefore, detection of gene expression abnormalities in cases of progressive growth is required. As in other areas of cancer research, the tumor microenvironment is coming to the forefront, also in vestibular schwannomas. In the paradigm of the tumor microenvironment, the stroma of the tumor actively influences the tumor’s behavior. However, research in the area of vestibular schwannomas is at an early stage. Thus, knowledge of the molecular mechanisms of tumorigenesis and interactions between cells present within the tumor is crucial for the diagnosis, prediction of tumor behavior, and targeted therapeutic interventions. In this review, we provide an overview of the current knowledge in the field of molecular biology and tumor microenvironment of vestibular schwannomas, as well as their relationship to tumor growth and hearing loss.

## 1. Introduction

Vestibular schwannoma (VS) is the most common tumor of the cerebellopontine angle. It is a benign tumor arising from the Schwann cells of the vestibular portion of the vestibulocochlear nerve. VSs grow from the peripheral zone of the superior or inferior vestibular nerve, lateral to the oligodendrocyte/Schwann cell junction (Obersteiner–Redlich zone, Figure 1), near the meatus, or within the internal auditory canal [1,2]. 

These tumors account for approximately 10% of all intracranial tumors and 80–95% of cerebellopontine angle tumors. The incidence of VS is around 10 to 15 per million per year, slightly higher among females than males. Most often, they manifest in the fourth and fifth decades of life [3]. While 95% of vestibular schwannomas (VSs) occur as unilateral sporadic tumors, the rest are associated with hereditary tumor syndrome neurofibromatosis type 2 (NF2), a germline mutation of the *NF2* gene, and rarely, VSs appear in association with inherited mutations of other tumor suppressor genes. In cases of NF2, bilateral VS are frequently seen [4]. Early signs of VS include sensorineural hearing loss, tinnitus, and vestibular symptoms. In the case of further growth, cerebellar and brainstem symptoms may be present together with other cranial nerve palsies. The diagnosis is suspected based on clinical symptoms, audiometric testing, and finally magnetic resonance imaging. The percentage of growing tumors varies widely between studies, ranging from 15 to 85%, probably due to the inconsistent definitions of tumor growth, measurement techniques, and length of follow-up. However, most tumors show an indolent growth pattern, i.e., there is no growth or very slow growth rates during follow-up [5,6,7]. 

Hearing loss is often the initial symptom of VS and is present in approximately 80–95% of patients during follow-up. Serviceable hearing is present in about 50% of patients at the time of diagnosis. Typically, the development of hearing impairment is slow in progression; however, cases of sudden deterioration have been described [8,9]. The etiology of the hearing loss associated with VSs remains unclear. Traditionally, direct compression of the cochlear nerve and restriction of the blood supply to the inner ear during tumor growth was believed to be the cause. However, hearing loss is also observed in non-growing tumors. Therefore, the cause of hearing loss is likely multifactorial, with the possible involvement of paracrine tumor activity. 

There are three options in the therapeutic approach to patients with VS: observation (wait and scan), radiotherapy/stereotactic radiosurgery, and microsurgical resection [10]. Except bevacizumab, no targeted therapy is available [11]. Due to the unpredictability of the symptom development and tumor growth, no clear consensus exists on the optimal treatment modality. Tumor size, hearing and facial nerve status, vestibular symptoms, patient age, comorbidities, and preference all play a role in the indication for an active approach [12]. Furthermore, the causes of VS tumorigenesis have not yet been clearly explained. Biallelic loss of the *NF2* tumor suppressor gene does not seem to be the only explanation. The tumor microenvironment and its influence on tumor behavior have received increasing interest in cancer research. However, a minimum of studies focuses on the tumor microenvironment of VSs. Pathophysiological mechanisms at the cellular and molecular levels allow a comprehensive understanding of tumor biology and open the door to diagnostic improvement and targeted therapeutic interventions.

## 2. Methods

For the purpose of this review, a search of the published literature in the PubMed database between January 2000 and July 2022 was conducted. Articles relating to the tumor biology and microenvironment of VS separately were included. Moreover, articles related to the tumor microenvironment with relation to tumor growth, patient hearing impairment, and targeted therapy were included. The search was conducted using the keywords “vestibular schwannoma”, “acoustic neurinoma”, “RNA analysis”, “single-cell RNA sequenation”, “tumor microenvironment”, “tumor growth”, “hearing loss”, “hearing impairment”. The inclusion criteria involved literature published between January 2000 and July 2022, English literature, and availability of full-texts. The exclusion criteria involved case reports, editorial notes and letters, conference abstracts, unavailable full-texts, and non-English literature. 

## 3. Peripheral Nerve Repair and Schwannoma Development

Schwann cells (SCs) contribute to the homeostasis and function of peripheral nerves. They originate from the ectomesenchyme of the neural crest. SCs can be divided into myelinating and non-myelinating SCs. However, more detailed analytical methods have identified additional subtypes of SCs, such as immature SCs, promyelinating SCs, and repair SCs [13,14]. They insulate neural fibers and play a fundamental role in nerve repair in cases of injury. Beyond SCs, other cell populations, such as immune cells, fibroblasts, and endothelial cells, also participate in peripheral nerve repair. 

When a peripheral nerve is injured, Schwann cells begin to express pro-inflammatory factors that recruit immune cells such as macrophages, neutrophils, and T lymphocytes to the site of the damage. Inflammation is involved in standard nerve degeneration processes [15,16] Myelin, located distally to the site of peripheral nerve injury, is fragmented and taken up by Schwann cells, later also by macrophages. Schwann cells remove myelin by autophagy and receptor-mediated phagocytosis. These processes initiate their transformation into the “repair cell” mode, which is essential for subsequent nerve regeneration [17,18,19]. This pathway, leading to regeneration of the peripheral nerve, is characterized by the high proliferative and migratory activity of the SCs, as well as by the production of vascular endothelial growth factor (VEGF) to promote new blood vessel formation [20]. Schwann cells then serve as a reference point for axon regeneration [21]. Regeneration is completed when signaling molecules, expressed by macrophages and regenerating axons (Neuregulin 1 type III-epidermal growth factor receptor ErbB2 pathway), induce redifferentiation of Schwann cells into a “quiescent” mode. A component of the new myelin sheath, myelin-associated protein, terminates the activity of macrophages and leads to their outflow from the site of injury [22,23,24]. 

Histopathologically, two types of areas are described in schwannoma tissue, Antoni A and Antoni B. The cell-rich Antoni A pattern consists of areas with densely ordered elongated nuclei forming “palisades” and non-cellular areas called Verocay bodies. The Antoni B pattern is disorganized, the stroma is loose, tumor cells have a more voluminous cytoplasm containing many lysosomes, and immune cells (monocytes, macrophages, lymphocytes) are abundant (Figure 2) [25]. It is hypothesized that Antoni B arises from degeneration of the Antoni A tissue arrangement during tumor progression. There are transition areas in the tumor with increased proliferative activity and a higher presence of infiltrating macrophages compared to the pronounced Antoni patterns [26]. In the past, similarities between the Antoni B pattern and the histopathological picture of Wallerian degeneration had been described [27].

Inactivation of tumor suppressor gene *NF2* located on chromosome 22q12, or loss of its functional product, the moesin–ezrin–radixin-like protein (merlin), plays a key role in the development of VS. According to Knudson’s double-hit hypothesis, this is a consequence of the loss of both functional alleles. NF2 patients carry a germline mutation in one allele of the *NF2* gene, and the development of VS is initiated after the inactivation of the second allele, either by deletion or mutation in the *NF2* locus or mutation of its regulatory regions [28]. Epigenetic modifications such as DNA methylation in the *NF2* gene promoter may also contribute to the reduced expression of this gene. In the case of sporadic schwannomas, these events occur independently in both alleles of the *NF2* gene, leading to somatic biallelic inactivation [28]. 

The merlin protein belongs to a group of ezrin–radixin–moesin proteins involved in cell remodeling and binds to the actin cytoskeleton near the cell membrane. Merlin is present in the cell in a dephosphorylated form, characterized by a folded conformation. In this conformational state, it acts as an active tumor suppressor. The unfolded dephosphorylated form of merlin binds to actin and stabilizes the attachment of the cytoskeleton to the cell membrane [29,30]. The main role of merlin as a tumor suppressor protein is to promote contact inhibition and suppression or inactivation of receptor-dependent promitogenic pathways. This is achieved by the association of merlin with membrane receptors. Contact inhibition is mediated by the interaction of merlin and hyaluronic acid transmembrane receptor CD44, which serves as a monitor of cellular density in the environment [31]. Furthermore, merlin influences signaling pathways such as Ras/Rac, Hippo pathway downregulating Yes-associated protein (Hippo-YAP), or mammalian target of rapamycin complex 1 (mTORC1) pathway, and affects the expression of growth factor receptor genes such as epidermal growth factor receptor (*EGFR)* gene [24,32,33]. Evidence supporting the relationship of merlin to these signaling pathways is shown by their overexpression upon merlin inactivation. 

Helbing et al. link the effect of *NF2* gene inactivation with subsequent upregulation of proliferation-promoting genes, to the impairment of effective peripheral nerve regeneration after mechanical stress, using the term “failure of nerve regeneration” [24]. This theory is supported by the predictive occurrence of schwannomas in anatomically exposed areas. These areas include the site where the nerve passes through or comes into contact with blood vessels or bony, muscular, and ligamentous structures. [24,34]. 

Schulz et al. investigated the effect of the loss of active *Nf2* gene alleles in vivo, using animal models. Genetically engineered mice with a selective knockout of the *Nf2* gene in Schwann cells (P0-Cre;Nf2) and/or neurons (Nefh-Cre;Nf2) were subjected to a sciatic nerve crush injury. After eight months, significant swelling of the injured nerve occurred in 90% of the animals with a homozygous knockout of the *Nf2* gene in Schwann cells (P0-Cre;Nf2^fl/fl^) and combined heterozygous knockout of the *Nf2* gene in Schwann cells and neurons (P0-Cre, Nefh-Cre;Nf2^fl/+^). Furthermore, in the case of combined heterozygotes, areas of the Antoni A schwannoma pattern were observed in the swelling of the injured nerve, but not in animals with a homozygous knockout in Schwann cells [35]. Previous findings of this research group show the influence of neuronal merlin on the expression of receptors and intracellular signaling of adjacent Schwann cells by affecting the Neuregulin 1 type III-ErbB2 signaling pathway. A decrease in the expression of Neuregulin 1 type III (Nrg1) was observed on the axonal surface in animal models with deletion of the neuronal *Nf2* gene and peripheral nerve biopsies of NF2 patients. In contrast, increased expression of ErbB2, the Nrg1 receptor, was also observed on the surface of Schwann cells in both groups [36]. Based on these results, the authors open the question of whether the development of schwannomas depends on the biallelic inactivation of the *NF2* tumor suppressor gene and suggest that intercellular interactions, together with the components of the tumor microenvironment, are important for the maintenance of tissue homeostasis and eventual tumorigenesis. 

## 4. Involvement of Other Genes in the Biology of VS

Several studies have investigated the presence and type of inactivation of the *NF2* gene in sporadic VS and concluded that at least one mutant allele is present in 49% to 100% of cases, with an average of 70%. Importantly, epigenetic modifications of the *NF2* locus or its regulatory regions might further affect *NF2* gene expression [37,38]. Among germline mutations, the leucine zipper-like transcription regulator (*LZTR1),* SWI/SNF related, matrix-associated, actin-dependent regulator of chromatin, subfamily B, member 1 *(SMARCB1)*, and protein kinase cAMP-dependent type I regulatory subunit alpha (*PRKAR1A)* genes have been reported in association with schwannoma development (schwannomatosis) [4]. Welling et al. reported upregulation of 42 genes in five of seven tumors examined and significant downregulation of eight genes in cDNA microarray analysis. Among the genes involved in cell proliferation and signaling, they highlighted the upregulation of genes encoding osteonectin (marker of decreased cell adhesion, angiogenetic mediator), endoglin (transforming growth factor β (TGF-β) receptor, a marker of angiogenesis), Ras homologous protein B (RhoB) GTPase (involved in cell signaling), and downregulation of lung cancer tumor suppressor gene *LUCA-15* (related to apoptosis) [39]. Cayé-Thomasen et al. applied microarray analysis to a group of 16 patients. Out of 111 upregulated genes involved in the cell cycle, differentiation, regulation of cell death, adhesion and extracellular matrix formation, and protein binding, they emphasized overexpression of platelet-derived growth factor D (PDGF-D), involved in the development of malignancies of the brain and other organ systems [38]. Aarhus et al. analyzed 25 sporadic VS. The main findings were the downregulation of the tumor suppressor gene encoding caveolin 1 (CAV1) protein and, in agreement with Welling et al., increased osteonectin expression. Other findings include deregulations of genes mostly related to the extracellular signal-regulated Ras-Raf-MEK-ERK signaling pathways [40].

More recently, a whole-genome analysis of 31 sporadic VS demonstrated increased activity of the hepatocyte growth factor/c-MET tyrosin kinase receptor (HGF/c-MET) pathway by altering the modulator expression, downregulation of the androgen receptor, and upregulation of the osteopontin gene involved in merlin degradation. The results of this study were not related to the mutational status of the *NF2* alleles [41].

As mentioned, noncoding parts of the genome, such as micro RNAs (miRNAs), and epigenetic mechanisms are also involved in gene expression. DNA methylation of gene promoters causes transcriptional silencing of tumor suppressor genes and participates in cancer development [42,43]. Gonzalez-Gomez et al. evaluated, in VSs, the DNA methylation status of 12 genes involved in cancer biology whose function is often suppressed by methylation in other types of cancer. Methylation was observed in nine genes studied: thrombospondin 1 gene *(THBS1),* tumor protein 73 gene *(TP73),* O-6-methylguanine-DNA methyltransferase gene *(MGMT),* merlin gene *(NF2),* tissue inhibitor of metalloproteinases 3 gene *(TIMP-3),* tumor protein 16 gene *(TP16),* caspase 8 gene *(CASP8),* retinoblastoma-associated protein 1 gene *(RB1),* and death-associated protein kinase gene (*DAPK)*. These genes are involved in the regulation of angiogenesis, DNA repair, inhibition of cell growth, apoptosis, and contact with the extracellular matrix (ECM) [44]. Similarly, Lassaletta et al. examined the methylation status of 16 tumor-associated genes in 22 VSs. In addition to most of the genes described by Gonzalez-Gomez et al., methylation was also observed in the genes coding Ras association domain family member 1A (*RASSF1A),* von Hippel–Lindau tumor suppressor *(VHL),* phosphatase and tensin homolog (*PTEN),* mismatch repair human mutL homolog 1 gene (*HMLH1),* and glutathione S-transferase P1 gene (*GSTP1)* [45,46]. Ahmed et al. described, for the first time in schwannomas, methylation of the gene promoter for apoptosis-associated speck-like protein containing a caspase recruitment domain (ASC) in 80% of the cases examined. In a human xenograft schwannoma model, they also showed that increased expression of ASC in tumor cells by viral gene transfection leads to slowing down the tumor growth, with a decrease in the Ki-67 and mitotic index and an increase in tumor cell apoptosis [47].

The genetic profile of sporadic schwannomas was performed by whole-exome sequencing by Havik et al. in a group of 46 tumors. Reference DNA was extracted from the peripheral blood cells of the patients. An average of 14 tumor-specific mutations were detected in each sample. A total of 692 gene mutations were detected, but the mutations were very heterogeneous. Only three genes were mutated in more than two tumor samples-*CDC27* (part of the anaphase-promoting complex, tumor suppressor), *USP8* (deubiquitinase gene, cell cycle involvement), and *NF2*. In the group of patients with intact *NF2* gene, mutations affecting SCs proliferation and *NF2* expression were the most frequent [48]. The most recent study in the field of whole-genome sequencing is the work of Shi et al., who reports changes in the expression of mitogen-activated protein kinase 8 interacting protein 1 (MAPK8IP1), solute carrier 36A2 (SLC36A2), and olfactory receptor 2AT4 (OR2AT4) [49].

## 5. First Proteomic Analyses of VSs

Several pioneering proteomic studies have focused on gene expression at the translational level. The advantage of this method is the evaluation of the final product of gene expression, as not all mRNA molecules detected at the transcription level are destined for translation. In addition, the final form of the protein is also affected by, e.g., post-translational modifications. In the first study, Seo et al. analyzed the proteomic profile of two VSs and two vestibular nerves. Twenty-nine differentially expressed proteins were identified, seven of them are associated with apoptosis-upregulated Annexin V, Annexin A4, Annexin A2 isoform 2, Tyrosine 3-monooxygenase/Tryptophan 5-monooxygenase activation protein zeta (YWHAZ), Rho GDP dissociation inhibitor alpha (ARHGDIA), heat shock protein 27, and downregulated peroxiredoxin 6. The upregulated genes were associated with both acceleration and suppression of apoptosis. The authors believe that this conflict could explain the slow growth of some VSs. In addition, these results support the concept of the Antoni B pattern that arises from the degeneration of Antoni A [50]. Xu et al. identified differential protein expression in 12 VSs and 12 normal vestibular nerves obtained from the same patient, presumably ipsilaterally, but there are no details about the sampling method of the normal nerve. It is arguable whether this control is appropriate given the assumption of the local mechanical and biological activity of the tumor. In general, choosing a proper control tissue for VS is difficult. An optimal control would be an intact vestibular nerve; however, except vestibular neurectomy in Meniere’s disease, the vestibular nerve is difficult to obtain [51]. Therefore, the great auricular nerve or other peripheral nerves have emerged as an alternative control in the studies [52,53]. The use of different control tissue across scientific studies reduces the ability to compare the results of these studies. Of numerous differentially expressed proteins, six significantly upregulated proteins were identified—lectin galactoside binding soluble 1 (LGALS1), Annexin 1, Annexin 2, growth factor receptor-bound protein 2 (GRB2), signal transducer and activator of transcription 1 (STAT1) and osteonectin, and one protein significantly downregulated—CAV1. The analysis did not show significant differences in merlin protein expression. Therefore, the authors believe that the role of merlin in the tumorigenesis of sporadic VS is limited [54].

## 6. First Single-Cell RNA Sequencing of VS Tumor Cells

The first cell-specific analysis of gene expression by single-cell RNA sequencing was performed in VS of three patients by Xu et al. Dysregulation in pathways related to proliferation, angiogenesis, energy metabolism, and immune response were detected. In the first clustering, two groups of SCs, large-cell SCs and small-cell SCs, were observed. In a more detailed analysis, five subpopulations were identified. Out of the five subpopulations, four subpopulations were found in the large-cell SC group. Each subpopulation was characterized by the expression of a specific gene: GDNF family receptor alpha 3 (GFRA3+), peripheral myelin protein 2 (PMP2^(high)^), FosB proto-oncogene (FOSB+), vascular endothelial growth factor A (VEGFA+), and periaxin (PRX+). PRX+ SCs, a single cluster of small-cell SCs, expressed markers of mature myelinating SCs and promyelinating SCs. Other subtypes showed similarity to non-myelinating, immature, and repair SCs and are thought to be the neoplastic cells. In these subgroups, a higher fraction of cells was found to be in the S phase of the cell cycle. The gene expression of each subgroup suggested the possibility of their specific roles in tumor biology, and candidate genes associated with the transformation of mature SCs into tumor cells were identified. *NF2* mutations were confirmed in all samples. However, the expression of merlin-regulated genes differed between samples, suggesting that different *NF2* mutations may have different pathogenicity [14].

## 7. Tumor Microenvironment

To better understand tumor biology, the area of interest in tumor research has broadened in the past two decades, from the tumor cells themselves to the tumor stroma and communication between cells present within the tumor mass. Tumors have come to be perceived comprehensively as a tumor microenvironment where all the components are fully integrated into the tumor’s biology. The tumor microenvironment is composed of the tumor cells themselves and the cellular and non-cellular components of the tumor stroma. The key effect of the non-tumor cell population, found within the neoplasm, has been described in many malignant and benign tumors. The cellular component of the tumor microenvironment can be a compound of immune cells, fibroblasts, platelets, stem cells, and other cell types. The non-cellular component involves structural proteins of ECM, regulatory proteins, and communication molecules such as cytokines, chemokines, and growth factors. In previous studies by our group, the key properties of the tumor microenvironment of the head and neck squamous cell carcinomas were described. We contributed to a better understanding of the function of cancer-associated fibroblasts and the theory of cancer stem cells [55,56,57,58,59].

## 8. Cellular Components of the VS Stroma

Until recently, research on the tumor microenvironment of VS has mainly focused on immune cells. The presence of an immune infiltrate has mainly been described in the Antoni B pattern [25]. In the context of tumor biology and behavior, macrophage infiltration is most often mentioned. Tumor-associated macrophages (TAMs) play a key role in the biology of many tumors, promoting tumor cell proliferation, neoangiogenesis, changes in the extracellular matrix, and influencing adaptive immunity. Monocytes enter the tumor from the peripheral circulation based on the chemotactic effects of tumor cells and their stroma. Subsequently, they differentiate into mature macrophages under the influence of the present cytokines. They are commonly differentiated into two categories, pro-inflammatory M1 and anti-inflammatory M2 macrophages. While M1 macrophages are actively involved in the management of infection, have antitumor activity, produce large amounts of inflammatory cytokines, and are good antigen-presenting cells, M2 macrophages control inflammation, are scavengers of dead cells, and promote angiogenesis and repair processes. In the case of TAMs, polarization to M2 occurs preferentially due to the composition of local effectors within the tumor tissue. TAMs promote tumorigenesis by secreting effectors such as VEGF, PDGF, tumor necrosis factor TNF-α, metalloproteinases (MMPs), and other molecules [60,61,62].

Schulz et al. assessed macrophage densities in histological sections of the injured nerve in the aforementioned mouse model study with a knockout Nf2 gene. As expected, only in (P0-Cre, Nf2^fl/fl^) homozygotes and combined (P0-Cre, Nefh-Cre; Nf2^fl/+^) heterozygotes were the macrophages significantly present. Staining of Arginase-1, a marker of the M2 macrophage subtype, suggested strong immunoreactivity in both model groups. The presence of macrophages and the M2 subtype was also analyzed in 40 human VS samples, demonstrating their presence in most sporadic and NF2-associated VS [35].

The first more comprehensive analysis of the immune microenvironment in VS was performed by Shi et al., who analyzed gene expression data from 46 sporadic VS and 11 controls by applying the CIBERSORT algorithm. This method allows the quantification of cellular fractions in tissue samples [63]. The tumor tissue displayed a significantly higher proportion of CD8+ T cells, CD4+ memory resting T cells, follicular helper T cells, activated NK cells, monocytes, activated dendritic cells, and eosinophils in comparison to control tissue. In contrast, the proportions of memory B cells, plasma cells, M2 macrophages, and resting mast cells were lower [49].

The first analysis of the complete cellular composition of VS is provided by the above-mentioned single-cell RNA sequencing study by Xu et al. In all three samples, seven components of the cellular stroma were identified based on the marker genes: microglia, fibroblasts, vascular smooth muscle cells, endothelial cells, proliferating microglia, T cells, and neutrophils. Although these cell types were present in the three patients, the representation of cells in each cluster varied significantly. For example, microglia accounted for more than 50% of cells in two patients, whereas in the third patient, it was only 25%. In contrast, SCs were represented in an inverse proportion to the microglia population, and other cell types showed variable representation between the patients. Analysis of intercellular communication identified a specific contact between fibroblasts and SCs mediated by insulin-like growth factors 1 and 2, and midkine growth factor, whose role in tumor pathophysiology has been described in many malignancies [14,64,65]. In contrast to other studies, macrophages are not present in the immune infiltrate, but the presence of microglia is massive. Microglia are cells of the myeloid lineage that originate in the yolk sac of the embryo. They migrate into the CNS during embryogenesis and remain there throughout life. There are different views on the renewal of microglia. While some authors favor exclusive in situ self-renewal, others admit the renewal of microglia from peripheral blood monocytes. A study investigating the convergence between microglia and peripheral macrophages during neuroinfections describes the expression of microglia-specific markers in CNS-infiltrating macrophages but not in other peripheral myeloid cells. Similar to macrophages, microglia can specialize into M1 and M2 phenotypes. Therefore, high plasticity of myeloid lineage cells is expected. The phenotype is influenced by local signaling molecules; thereby, surface markers lose the ability to distinguish these cells clearly [66,67,68].

## 9. Non-Cellular Components of the VS Stroma

In addition to cellular components, cytokines, chemokines, and growth factors are also involved in the biology of VS. Taurone et al. presented increased expression of pro-inflammatory cytokines such as transforming growth factor β1 (TGF-β1), interleukin 1β (IL-1β), interleukin 6 (IL-6), and TNF-α in tumor tissue [51]. Among the C-X-C motif chemokine family, increased expression of CXCL16 and its receptor CXCR6/Bonzo has been described. This ligand–receptor pair promotes cell growth and migration in tumor-culture experiments [69]. Furthermore, increased expression of CXCR4, a receptor that plays a role in the homing of stem cells, progenitor cells, and immune cells, has been described in VS. In malignant neoplasms, CXCR4 is involved in the acceleration of tumor growth and invasiveness [53]. 

The humoral component of the tumor microenvironment is related to the expression of the cyclooxygenase 2 enzyme, which has gained growing interest in recent years. Cyclooxygenase (COX) metabolizes arachidonic acid into prostaglandins, prostacyclins, and thromboxane. It exists in two isoforms, constitutive COX1, and inducible COX2, associated with the inflammatory response. COX2 is discussed in the context of tumor biology mainly because of its association with tumor progression and the possibility of therapeutic targeting by commonly available drugs such as aspirin and other NSAIDs. Hong et al. described a positive correlation between COX2 and the expression of proliferation marker Ki-67 in VS in a group of 15 sporadic VS and 15 NF2-associated VS [70,71]. 

Of the growth factors, Taurone et al. described the increased expression of VEGF in schwannoma tissue, a growth factor associated with neoangiogenesis in many tumors [51]. The presence of VEGF was described in 100% of tumors by Plotkin et al. [72]. Lewis et al. described a close relationship between neovascularization and inflammatory infiltrates in their study comparing the microenvironment of sporadic and NF2-associated schwannomas. The mediator of this relationship was VEGF and its receptor VEGFR-1. The source of the growth factor and its receptor were TAMs. In addition to induction of angiogenesis, promotion of vasodilation, and increased vascular permeability, VEGF acts as a chemoattractant for circulating monocytes expressing VEGFR-1. VEGF also acts as a polarizer of monocytes to the M2 phenotype [73].

A schematic presentation of the VS tumor microenvironment is shown in Figure 2 (Figure 3).

## 10. Relation of the VS Biology and Microenvironment to Tumor Growth

### 10.1. NF2 and Other Modifications in Gene Expression

The relationship between the specific mutation of the *NF2* gene and the effect of the inactivation of both *NF2* alleles on tumor growth has not yet been described in VSs. Sass et al. described the increased expression of genes for Erbin (ErbB2 interacting protein), platelet-derived growth factor C (PDGF C), phosphatidylinositide 3-kinase (PI3K), α-actinin 1, and toll-like receptors in fast-growing sporadic VS, while brain-specific protein and neural cell adhesion molecule 1 (NCAM 1) were significantly downregulated [74].

### 10.2. Differences in the Cellular Tumor Microenvironment

With a focus directed toward the tumor microenvironment, tumor-associated macrophages are most often mentioned when discussing VS growth. In a group of 67 sporadic VSs, de Vries et al. described a positive correlation between CD45 and CD68 expression and tumor growth index (maximal tumor diameter divided by patient age). Furthermore, a positive correlation was observed between CD68+ cell count and microvascular density [75]. One year later, these authors described a positive relationship between CD163 expression (M2 macrophage marker), microvascular denseness, and tumor growth [76]. Lewis et al. observed more extensive infiltration of Iba1+ macrophages in growing tumors compared to non-growing tumors. Moreover, macrophages were the most abundant population in the growing tumors (they were present in 50–70% of cases) [77]. Perry and Graffeo described the higher density of macrophages in a group of patients with tumor progression after subtotal resection. In contrast to the results presented by de Vries et al., they observed an increased number of M1-polarized macrophages [78,79]. Goncalves et al. performed an extensive immunohistochemical study in 923 tumor samples. The lymphocyte and macrophage infiltrate were assessed by expression of CD3, CD8, CD68, and CD163. A positive correlation was found between increased marker expression and tumor size, but no relationship with volumetric growth was evident. Furthermore, when macrophage and lymphocyte markers were assessed together, a higher density of these cells was surprisingly associated with slower tumor growth [80]. 

### 10.3. Differences in the Non-Cellular Tumor Microenvironment

De Vries et al. described a positive relationship between tumor expression of macrophage colony-stimulating factor (M-CSF) and rapid growth and cystic organization of VSs. Similarly, CD163 expression was increased in these tumors [81]. 

The relationship of COX2 expression, an indirect marker of inflammatory cytokine production, to preoperative volume and volumetric growth in a cohort of 898 patients was investigated by Behling et al. COX2 was upregulated in younger patients with larger tumors. However, no correlation was found between COX2 expression and tumor growth [82]. 

Cayé-Thomasen et al. described a positive correlation between the presence of VEGF and tumor growth, first in an immunohistochemical study, then by measuring VEGF and VEGFR-1 concentrations in VS tissue homogenates [83,84]. Indirect evidence of the association between VEGF and schwannoma growth is provided by the effect of bevacizumab, an anti-VEGF antibody used to treat patients with NF2. A reduction in tumor volume has been observed after its use [11]. Exploring the factors that influence the formation of ECM, Møller et al. observed a positive correlation between increased MMP-9 concentration in the tumor homogenates and absolute tumor growth rate [85]. 

The aforementioned studies are summarized in an overview table (Table 1).

## 11. A New Concept of Hearing Loss in Patients with VS

The hearing loss in VS patients was traditionally explained by direct compression and stretching of the cochlear nerve during tumor growth, associated with obliteration of its vascular supply. However, the onset of hearing loss is frequently unpredictable, an association with small tumors has been described, and the progression of hearing loss can be observed even in patients with non-growing tumors. In a current systematic review, Lassaletta et al. note the controversial results of studies that correlated the severity and progression of hearing loss with tumor size or growth [86,87,88]. Other studies have suggested cochlear abnormalities such as intralabyrinthine hemorrhage, obstruction of the cochlear aperture, endolymphatic hydrops, accumulation of intralabyrinthine protein, and paracrine ototoxic tumor activity as additional causative factors in the development of hearing loss [88,89,90,91]. In our recent unpublished results, the removal of VS with preservation of hearing led to normalization of the intensity of inner ear signal on magnetic resonance imaging, which should be in line with the change of protein concentration in the perilymph. However, 25% of the patients in the cohort developed hearing loss from useful to useless in postoperative follow-up. Interestingly, the 10-year prognosis for hearing preservation is the same between individual treatment strategies. On average, 25–50% of patients preserved their hearing function for 10 years after diagnosis, regardless of the procedure chosen. Thus, hearing is impaired even in patients with non-growing tumors who are included in the observational branch [92]. 

## 12. Relation of the Biology and Microenvironment of VS to Hearing Loss

### 12.1. NF2 and Other Modifications in Gene Expression 

A pioneering work that relates hearing loss to VS molecular processes was that of Lassaletta et al., who described a positive correlation between tumor protein P73 (*TP73)* gene methylation and hearing loss based on pure tone audiometry (PTA) [45]. One year later, these authors described a longer duration of deafness and higher 2000 Hz hearing thresholds in a patient with negative cyclin D1 immunostaining in the tumor samples [93]. In a whole-genome expression study, Stankovic et al. described the reduced expression of peroxisomal biogenic factor 5-like (PEX5L) associated with hearing loss. Decreased expression of PEX5L causes peroxisomal dysfunction resulting in pathological accumulation of fat tissue and/or demyelination and neurodegeneration of the acoustic portion of the vestibulocochlear nerve. Other genes, encoding RAD54 homolog B (RAD54B) and prostate-specific membrane antigen-like protein (PSMAL), are under-expressed in association with poor hearing. In contrast, an increase in the expression of carcinoembryonic antigen was observed [94]. Focusing on the *NF2* gene, Lassaletta et al. observed lower mean corrected PTA thresholds in patients with VS-carrying mutations in the *NF2* gene [95]. 

### 12.2. Differences in the Cellular Tumor Microenvironment 

Sagers et al. were the first to postulate the association between macrophages and hearing loss in VSs. They observed increased expression of genes essential for inflammasome formation in VS. The protein products NOD-, LRR-, and pyrin domain-containing protein 3 (NLRP3) and IL-1β, as well as macrophage marker CD68, were expressed dominantly in tumors from patients with hearing loss [52]. Recently, an immunohistochemical study of 30 tumors has found significantly higher expression of CD163 in tumors of poor hearing patients, suggesting a predominance of M2 macrophages. Furthermore, the total number of macrophages was slightly higher in patients with impaired hearing [8].

### 12.3. Differences in the Non-Cellular Tumor Microenvironment

Groundbreaking studies correlating hearing loss in VS with possible tumor ototoxic and otoprotective secretion were published in 2013 and 2015 by Dilwali et al. First, increased expression of the hearing-protective fibroblast growth factor 2 (FGF2) was found to be positively correlated with a good hearing in VS patients [96]. Later, TNF-α was identified as an ototoxic molecule produced by VSs, and its secretion levels were positively correlated with hearing loss. In a cochlear explant culture experiment, the addition of recombinant human TNF-α to the cultivation medium led to the loss and disorganization of neurites in the basal turn of the cochlea [97]. Another protein overexpressed in correlation with hearing impairment is the chemokine receptor CXCR4 described above [53]. As in tumor growth, the positive effect of targeted anti-VEGF therapy on hearing development in patients with NF2-related VS indirectly suggests that VEGF is associated with hearing impairment. However, no selective study of the relationship between VEGF and hearing in VS has been performed [72,98]. Another putative mechanism of hearing impairment caused by tumor secretory activity is the production of extracellular vesicles with the potential to destroy inner-ear hair cells. Only the vesicles obtained from the tumors of patients with poor hearing damaged both cochlear sensory cells and neurons [99]. Interestingly, a recent publication has reported that plasma concentrations of MMP-14 metalloproteinase correlate positively with the preoperative degree of sensorineural hearing loss in surgically treated patients. Direct MMP-14-induced neuronal cell damage was demonstrated in a cochlear explant model [100].

In general, the tumor microenvironment appears to play a key role in tumor-induced hearing loss in patients with VSs.

The aforementioned studies are summarized in an overview table (Table 2).

## 13. Clinical Application of the Knowledge of the Biology and Microenvironment of VS-Diagnosis and Targeted Therapy

The above findings are mainly based on analyses performed on VS tissue obtained from patients who have undergone surgical resection of VS. Currently, the first studies demonstrating the detection of prognostic markers in vivo are being published. In the field of imaging methods, Lewis et al. described the increased accumulation of a specific neuroinflammation PET tracer, ^11^C-(R)-PK11195, in patients with growing sporadic VS. This tracer binds to the so-called translocation protein (TPSO), which is abundantly expressed in immune cells. The authors described a spatial correspondence between TPSO and macrophage-specific marker ionized calcium-binding adapter 1 (Iba-1), suggesting that the increased signal in growing tumors originates from a higher percentage of macrophages in the tumor tissue. An increase in the K^trans^ parameter for vascular permeability was also observed in dynamic contrast-enhanced MRI in growing tumors, which was correlated with the two previous markers [77]. Another study showed in vivo CXCR4 expression imaging with ^68^Ga-Pentixafor-PET/CT. In the previous year, CXCR4 overexpression was described in patients with poor hearing. Therefore, this imaging method could theoretically serve as a detection method for patients at risk of hearing loss [53,101]. The first prospective predictive marker of tumor behavior in commonly available biological material appears to be the MMP-14 enzyme. As noted above, serum MMP-14 concentrations were positively correlated with the preoperative hearing status of patients [100].

With the development of analytic methods in molecular biology, the topic of targeted tumor therapy is coming to the forefront of antitumor research. Understanding the mechanism of tumor initiation and progression is a prerequisite for thinking about selectively influencing the metabolic pathways involved. In the case of VS, only the anti-VEGF monoclonal antibody bevacizumab targeting tumor neoangiogenesis has been introduced into clinical practice to date. However, its use is currently reserved for patients with progressive NF2-associated schwannomas. Several studies have described the effect of bevacizumab in terms of tumor shrinkage and improved hearing in patients. On the other hand, its side effects, such as gastrointestinal dysfunction, changes in the blood count, and liver and renal dysfunction, cannot be overlooked [11,72,102,103]. Currently, in Japan, the first randomized double-blind multicenter trial (BeatNF2) has been conducted to assess the effect and safety of bevacizumab use in NF2 patients. However, the results of this study are not yet available [104]. 

As described above, merlin inhibits pro-proliferative signaling mediated by receptor tyrosine kinases. In this regard, protein kinase inhibitors used in the targeted treatment of various neoplasias may serve as a “substitute” for merlin’s function. Currently, phase II clinical trials have been performed for crizotinib (c-MET kinase inhibitor and anaplastic lymphoma kinase inhibitor) and everolimus (mTORC1 inhibitor, derivative of rapamycin) in patients with NF2. Among cytokines, CXCR4, experimentally used as a radiotracer ligand for PET CT imaging, has potential in targeted therapy. A CXCR4 antagonist, plerixafor, could be used as a future receptor-targeted drug [105]. 

The effect of cyclooxygenase inhibitors, aspirin, and other NSAIDs on tumor behavior is controversial. Increased expression of the COX2 enzyme has been observed in VSs with increased proliferative activity and greater preoperative volume. To evaluate the effect of these agents on tumor behavior, patients with a history of aspirin use, primarily due to cardiovascular disease, were followed in cohorts. Kandathil et al. described an inverse association between aspirin intake and volumetric growth [106]. No correlation between tumor behavior and the use of anti-inflammatory medications has been found in other studies [107]. The cause may be an insufficient dose. However, the in vivo dosage needs are unknown. Currently, to answer this question, a prospective randomized double-blind study (NCT03079999) is being designed to determine the effect of aspirin on tumor growth and hearing development in patients affected by VS. The proposed dose is 325 mg twice a day [82]. 

## 14. Conclusions

Approaching tumors as a complex tumor microenvironment with the active involvement of all its cellular and non-cellular components is crucial to understanding their behavior. Developing methods of molecular biology are helpful in elucidating the probable causes of tumorigenesis and open the way to improved diagnostics, the establishment of prognostic markers, and targeted antitumor therapy. This review provides an overview of the current knowledge of VS from this perspective. 

The majority of the reported transcriptomic and proteomic studies provide a larger amount of information by processing the tumor tissue as a bulk, which is a very low-resolution approach in terms of the concept of the tumor microenvironment. Cellular specification of gene expression is impossible using the abovementioned methods. From this point of view, single-cell assessment methods such as single-cell RNA sequencing and spatially resolved transcriptomics in tissue sections are optimal. Therefore, we are still in the early stages of understanding the tumor microenvironment, and further research is required to decipher the key mechanisms of VS formation and behavior at the cellular and subcellular levels.

A future perspective lies in the identification of markers that can especially predict the prognosis of tumor growth and hearing impairment. These markers should be monitored by a minimally invasive method, e.g., in the cerebrospinal fluid or plasma of affected patients. Tumor growth progression and hearing loss are some of the most important factors that influence decision-making in choosing the treatment strategy. Understanding the mechanisms of these processes could modify indications for surgical treatment. Furthermore, knowledge of the biological profile of the tumor could provide new options in the treatment strategy directed toward targeted therapy. 

## Figures and Tables

**Figure 1 biomedicines-11-00032-f001:**
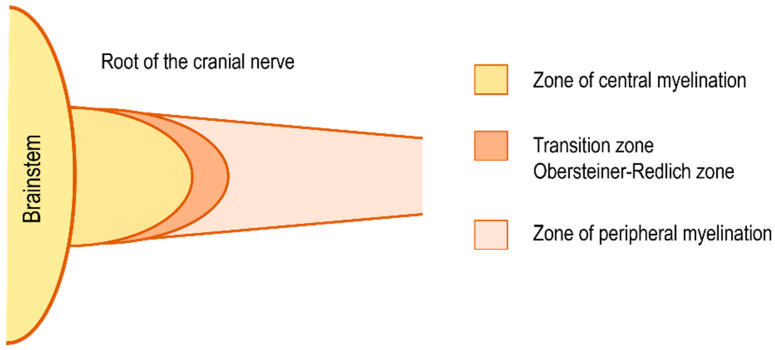
Scheme of the transition zone (Obersteiner–Redlich zone) between the central and peripheral zone of myelination of the cranial nerve. Zone of peripheral myelination lateral, centrifugal, to the transition zone is the site of vestibular schwannoma origin.

**Figure 2 biomedicines-11-00032-f002:**
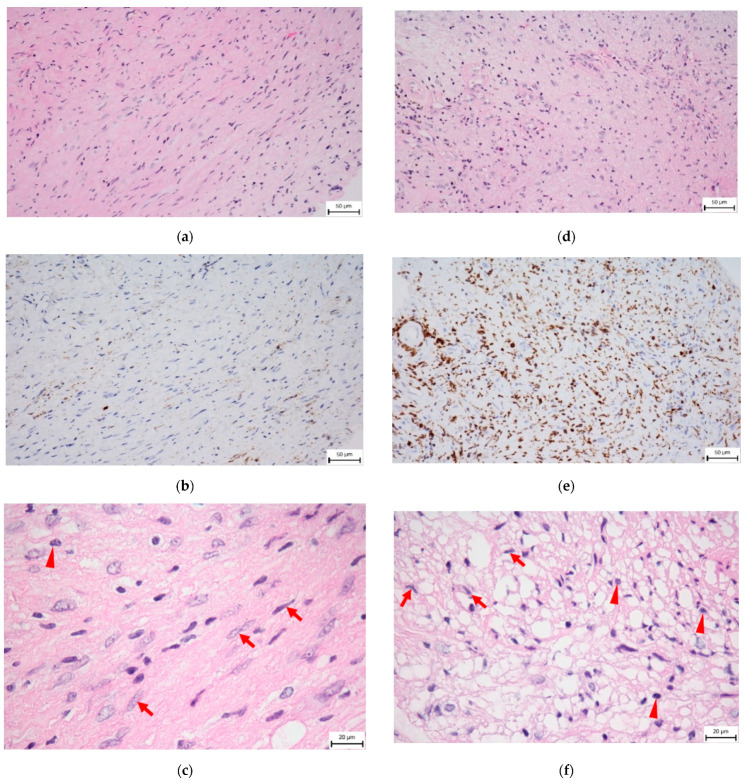
Histology of VS, sections from the same tumor. Arrows show tumor cell nuclei; triangles point to macrophage nuclei. (**a**) Hematoxylin–eosin staining of tumor cell-rich Antoni A area (400× magnification). High density of tumor cells, presence of elongated cell nuclei. (**b**) CD163 immunohistochemical staining of Antoni A area (400× magnification), low CD163^+^ macrophage density (brown). (**c**) Hematoxylin–eosin staining-Antoni A area (1000× magnification). (**d**) Hematoxylin–eosin staining of Antoni B area (400× magnification). The disorganization of cellular arrangement, presence of microcystic pattern, and different shapes of cell nuclei. The stroma is loose. Immune cells are abundant. (**e**) CD163 immunohistochemical staining of Antoni B area (400× magnification), high CD163^+^ macrophage density (brown). (**f**) Hematoxylin–eosin staining-Antoni B area (1000× magnification). Bar is 50 μm (**a**,**b**,**d**,**e**) and 20 μm (**c**,**f**). This figure was provided by the Department of Pathology and Molecular Medicine, 2nd Faculty of Medicine, Charles University and Motol University Hospital.

**Figure 3 biomedicines-11-00032-f003:**
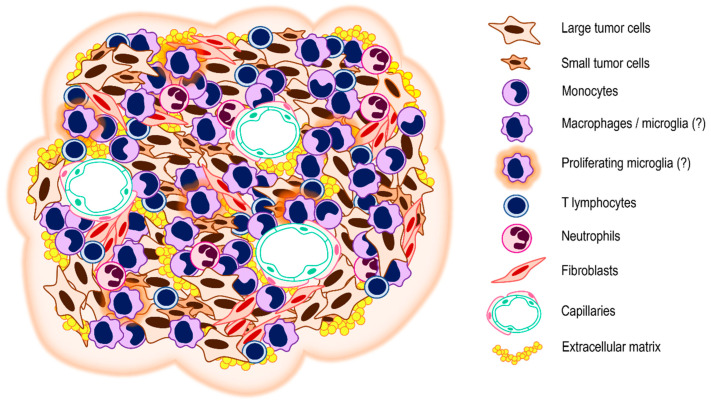
Schema of the VS tumor microenvironment. The tumor microenvironment is composed of the tumor cells themselves and the cellular and non-cellular components of the tumor stroma. The cellular component of the tumor microenvironment of VS is a compound of immune cells and mesenchymal cells. The non-cellular component involves structural proteins of the extracellular matrix, regulatory proteins, and communication molecules such as cytokines, chemokines, and growth factors.

**Table 1 biomedicines-11-00032-t001:** Relation of the VS biology and microenvironment to tumor growth—literature overview. (sVS—sporadic VS, STR—stereotactic radiosurgery, PDGF—platelet-derived growth factor, PI3K—phosphatidylinositol 3-kinase, NCAM—neural cell adhesion molecule, IHC—immunohistochemistry staining, Iba-1—ionized calcium-binding adaptor molecule 1, TAMs—tumor-associated macrophages, VEGF—vascular endothelial growth factor, VEGFR—vascular endothelial growth factor receptor, MMP—matrix metalloproteinase, TIMP—tissue inhibitor of metalloproteinases, M-CSF—macrophage colony-stimulating factor, IL-34—interleukin 34, COX—cyclooxygenase).

Author	Year	Number of Patients	Analytical Method	Conclusions
*NF2* and other modifications in gene expression				
Sass et al. [74]	2017	16 sVS	Microarray analysis	▪ Increased expression of Erbin, PDGF C, PI3K, α-actinin 1, toll-like receptors in fast-growing tumors ▪ Decreased expression of brain-specific protein and NCAM 1 in fast-growing tumors
Differences in the cellular tumor microenvironment				
De Vries et al. [75]	2012	67 sVS	CD45, CD68 (IHC)	▪ CD45 and CD68 expression positively correlated with tumor growth index, tumor size, and microvessel density
De Vries et al. [76]	2013	10 slow-growing sVS 10 fast-growing sVS	CD163 (IHC)	▪ Significantly higher CD163 and microvessel density in fast-growing tumors compared to the slow-growing cohort
Graffeo et al. [78]	2018	46 sVS after STR 33 stable 13 progressive	CD68 (IHC)	▪ Significantly higher macrophage density was associated with progressive growth of the residual tumor
Lewis et al. [77]	2019	5 growing sVS 3 static sVS	Iba-1 (IHC)	▪ Significantly higher maximum and mean Iba-1+ macrophage count in the growing tumors ▪ Macrophage count predominance in growing tumors (50–70%)
Perry et al. [79]	2019	44 sVS after STR 32 stable 12 progressive	CD163 (IHC)	▪ M1 macrophage predominance in TAMs ▪ CD163 positivity and M2 index (CD163+ cell count divided by CD68+ cell count) were significantly increased in tumors that remained stable
Goncalves et al. [80]	2021	923 sVS volumetric growth analysis in 189 sVS	CD3, CD8, CD68, CD163 (IHC)	▪ Larger preoperative tumor size in tumors with increased levels of the assessed markers ▪ No correlation between assessed markers and volumetric growth ▪ Higher density of both macrophages and lymphocytes was associated with slower tumor growth
Differences in the non-cellular tumor microenvironment				
Cayé-Thomasen et al. [84]	2003	15 sVS	VEGF (IHC)	▪ The level of VEGF expression correlates positively with a tumor growth rate
Cayé-Thomasen et al. [83]	2005	27 sVS	VEGF, VEGFR-1 (concentrations)	▪ Significant positive correlation between the VEGF and VEGFR-1 concentration in tumor homogenates and tumor growth rate
Møller et al. [85]	2010	34 sVS	MMP-2, MMP-9, TIMP-1 (IHC and concentrations)	▪ Significant positive correlation between the MMP-9 concentration in tumor homogenates and absolute tumor growth rate, weak for the relative growth rate ▪ No correlation between MMP-2 and TIMP-1 and tumor growth or other clinical parameters
De Vries et al. [81]	2019	10 fast-growing sVS 10 slow-growing sVS	M-CSF, IL-34, CD163 (IHC)	▪ Higher expression of M-CSF in fast-growing and cystic tumors ▪ CD163 expression was higher in tumors with high M-CSF staining ▪ All tumors expressed IL-34, but no significant differences were found in relation to clinical parameters
Behling et al. [82]	2021	898 sVS volumetric growth analysis 171 sVS	COX2 (IHC)	▪ Larger preoperative tumor volume in younger patients and higher COX2 expression ▪ No association between COX2 expression and tumor growth

**Table 2 biomedicines-11-00032-t002:** Relationship of the biology and microenvironment of VS to hearing loss—literature overview. (sVS—sporadic VS; NF2-VS—VS associated with neurofibromatosis type 2; PCR—polymerase chain reaction; TP53—tumor protein p53; PEX5L—peroxisomal biogenic factor 5-like; RAD54B—RAD54 homolog B; PSMAL—prostate-specific membrane antigen-like protein; CEA—carcinoembryonic antigen; dHPLC—denaturing high-performance liquid chromatography; PTA—pure tone audiogram; *NF2*—merlin gene, tumor suppressor; GAN—great auricular nerve; qRT-PCR—quantitative real-time reverse transcription PCR; NLRP3—NOD-, LRR- and pyrin domain-containing protein 3; IL-1β—interleukin 1β; ELISA—enzyme-linked immunosorbent assay; FGF2—fibroblast growth factor 2; TNF-α—tumor necrosis factor α).

Author	Year	Number of Patients	Analytical Method	Conclusions
*NF2* and other modifications in gene expression				
Lassaletta et al. [46]	2006	21 sVS, 1 NF2-VS	Methylation-specific PCR	▪ Methylation of the *TP53* gene was associated with hearing loss
Lassaletta et al. [94]	2007	21 sVS	Cyclin D1 (IHC)	▪ Negative cyclin D1 expression in patients with longer duration of hearing loss and higher 2 kHz hearing thresholds
Stankovic et al. [95]	2009	9 sVS—good hearing 4 sVS—poor hearing	Microarray analysis	▪ Decreased expression of PEX5L, RAD54B, PSMAL, and increased expression of CEA in patients with poor hearing
Lassaletta et al. [96]	2013	51 sVS	PCR/dHPLC direct sequencing	▪ Lower mean corrected PTA thresholds in patients with *NF2* mutations
Differences in the cellular tumor microenvironment				
Sagers et al. [53]	2019	15 sVS—good hearing 15 sVS—poor hearing 7 GAN controls	qRT-PCR IL-1β, NLRP3, CD68 (IHC)	▪ Significant upregulation of NLRP3 inflammasome-associated genes in tumors of patients with poor hearing ▪ Increased IHC positivity for IL-1β, NLRP3, and CD68 in patients with poor hearing compared to well-hearing patients, without statistical significance
Nisenbaum et al. [8]	2021	12 sVS—good hearing 15 sVS—poor hearing 3 NF2-VS—poor hearing	CD80, CD163 (IHC)	▪ Higher expression of CD163 and slightly higher macrophage count in tumors of poorly hearing patients
Differences in the non-cellular tumor microenvironment				
Dilwali et al. [97]	2013	16 sVS—good hearing 19 sVS—poor hearing 7 GAN controls	Cytokine array ELISA of secretions	▪ FGF2 was 3.5-fold higher in tumors associated with good hearing (cytokine array) ▪ Higher levels of FGF2 in secretions of well-hearing patients
Dilwali et al. [98]	2015	3 sVS—good hearing 10 sVS—poor hearing 2 GAN controls	ELISA of secretions Murine cochlear explant culture	▪ TNF-α positively correlated with PTA and negatively with word recognition scores ▪ TNF-α led to damage and disorganization of neurites in the basal turn of cochlear explants
Soares et al. [100]	2016	3 sVS—good hearing 3 sVS—poor hearing	Extracellular vesicle analysis Murine cochlear explant culture	▪ Damage of cochlear cells and neurons by extracellular vesicles of tumors in patients with poor hearing
Ren et al. [101]	2020	4 sVS—good hearing 19 sVS—poor hearing 6 GAN controls	MMP-14 (IHC, activity in VS secretions, activity in plasma) Murine cochlear explant culture	▪ Plasma concentrations of MMP-14 correlated with the preoperative degree of hearing loss ▪ Neuronal cell damage in cochlear explant model caused by MMP-14
Breun et al. [102]	2018	10 NF2-VS—good hearing 8 NF2-VS—moderate hearing 12 NF2-VS—poor hearing 8 sVS—good hearing 14 sVS—moderate hearing 8 sVS—poor hearing	CXCR4 (mRNA-PCR, IHC, Western blotting)	▪ Overexpression in both sVS and NF2-VS group ▪ Higher CXCR4 expression in patients with hearing impairment

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
