# Peer review of "Tumor Biology and Microenvironment of Vestibular Schwannoma-Relation to Tumor Growth and Hearing Loss"

_biomedicines, 2022, doi:10.3390/biomedicines11010032_

Round 1

Reviewer 1 Report

A really nice update on the molecular bases of vestibular schwannomas. Just some remarks:

- please give some methodological criteria for this work: was a literature search conducted before? why some papers are discussed and others are not? is only the most recent literature included? etc.

- figure 1, give credits where the pictures come from;

- line 154, citation is missing;

- lines 482...: I personally do not like to introduce unpublished data in the middle of a review and if you want to keep it then you must give more details as to surgical techniques used, follow up time, detailed audiological results pre and post etc.

- what are the criteria for article selection in table 1 and 2: is there a time span for inclusion? 

- some tenses are clumsy, there are some little mistakes throughout the text (eg. 287 out of, 301 in order to, etc. etc.), and please do not mix past and present form of the verbs in the same sentence.

Reviewer 2 Report

The subject the authors attempt to tackle, VS pathophysiology and origins of VS associated hearing loss is of great importance. Given the subject’s importance and considerable work that has gone into preparing this manuscript so far, authors should be encouraged to correct this article's many deficiencies. English language use is generally quite good but their exposition is vague or inaccurate at times and must be corrected. The run-on paragraphs must be corrected. It isn’t English language editing this paper needs, It is scientific/medical exposition editing it needs. I have documented some of these inaccuracies below. I would welcome the chance to re-review this work after corrections.

Once vestibular schwannoma was defined - abbreviated - as VS, use VS thereafter in Abstract. Then define again in text and after first definition use VS thereafter too. Numerous other inconsistencies must be fixed [example - sometimes merlin starts with upper case sometimes lower case].

Line 39, after first mention of “vestibular schwannoma” abbreviate as VS there, then use VS thereafter.

Line 43. This is an important point deserving a figure or drawing showing the transition zone from Schwann cell myelin to oligodendrocyte myelin, the Redlich–Obersteiner zone. The authors must always think of their readers. Articles in a general journal like Biomedicines must not assume specialist knowledge. Also the Redlich–Obersteiner's zone has deeper pathophysiological meaning for VS development.

Line 51, “ bilateral impairment” is wrong. One can say “ bilateral hearing impairment” or “ bilateral vestibular function impairment” or “VS are found bilaterally in __% of cases”. 

Line 59, this may be my idiosyncratic view but breaking up run-on paragraphs to keep one theme per paragraph helps readers. Consider the matter of hearing at line 59 starting a new paragraph.

Line 68 must start a new paragraph. Not optional.

Line 76, this needs rephrasing. You have already established and stated that NF2 mutation is present in 5% of VS cases. Don’t repeat and don’t be vague [ re “some¨]. 

Lines 82-84. Why are you saying this ? isn;t it clear to anyone reading this ?

Line 85 is wrong. “SCs contribute to…” would be correct.

Line 90. Is wrong or misleading. Wallerian degeneration refers to axons that degenerate after trauma or toxic insult. As an aside I would like to see a few references to axonal degeneration after myelin only destruction or defects. It is generally assumed that such occurs but I haven’t seen direct experimental proof of this.

Line 106 to 110. Isn’t this wrong ? Wouldn’t you agree that regeneration is complete when status quo ante has been established ?

Fig 2 is excellent. I advise the authors to add several drawings [figures] depicting a VS and the transition zone. Re. Fig. 1 micrographs of Antoni A and Antoni B stating magnification important. It would not be usual, but I think it would be useful to add micrographs at high power oil views of both 1A and 1B micrographs. See Wippold et al for a good example of introducing VS physiology to non-specialists. Re. fig. 1 c, the fault could be mine but I do not see the cells mentioned in legend. I would also object to use of term “immune cells” in a scholarly article like this. What are they ? Doesn’t lymphocyte infiltration have little physiologically in common with neutrophil infiltration ? 

Re line 130, “Inactivation of tumor suppressor gene NF2 located on chromosome 22q12, or loss of its functional product, the moesin-ezrin-radixin-like protein (merlin), plays a key role in the development of vestibular schwannoma.” it is unclear if the authors are talking of people with NF2 mutation or VS in non-mutated ?

Line 137 must start a new paragraph.

Line 138, do not abbreviate “ezrin-radixin-moesin (ERM)”. You never again used the abbreviation.

Also keep in mind that the fewer abbreviations you use, the easier it will be for readers.

Lines around 140, Merlin is of such importance that it requires a diagram of its activation, downstream effects on Hippo, MTOR etc, and its interaction with actin. The details regarding this are important to the authors’ project.

Lines 154 to 162 are central to this authors’ paper. This paragraph is crucial. The paper must be rewritten to focus on making this point clear.

The fault could be mine but I have been unable to understand paragraph 164-183. Authors state “Based on these results, the authors open the question of whether the development of schwannomas depends on the biallelic inactivation of the NF2 tumor suppressor gene and suggest that intercellular interactions, together with the components of the tumor microenvironment, are important for the maintenance of tissue homeostasis and eventual tumorigenesis.” yet earlier in paragraph state “... in the case of combined heterozygotes, areas of the Antoni A schwannoma pattern were observed in the swelling of the injured nerve, but not in animals with homozygous knockout in Schwann cells [36].” Aren’t these sentences conflicting ?

I stopped reviewing at this point but would welcome reviewing the paper again after better organization, etc. See references below for suggestions on sound medical exposition practice.

---------------------------------------------------------------------------

Wippold FJ 2nd, Lubner M, Perrin RJ, Lämmle M, Perry A. Neuropathology for the neuroradiologist: Antoni A and Antoni B tissue patterns. AJNR Am J Neuroradiol. 2007 Oct;28(9):1633-8. doi: 10.3174/ajnr.A0682. 

1: Alvarez Aquino A, Ramirez MJE, Bozkurt I, Asprilla González JA, Goncharov E,

Caballero AD, Nurmukhametov R, Montemurro N, Chaurasia B. Treatment of

Intracranial Tumors With Stereotactic Radiosurgery: Short-Term Results From

Cuba. Cureus. 2022 Oct 5;14(10):e29955. doi: 10.7759/cureus.29955. PMID:

36348852; PMCID: PMC9635578.

2: Agarwal P, Natanasabapathi G, Bisht RK, Malhotra RK, Kale SS. Investigation

of optimal planning strategy in gamma knife perfexion for vestibular schwannoma

tumor using hybrid plan technique. Biomed Phys Eng Express. 2022 Nov 18;8(6).

doi: 10.1088/2057-1976/ac9abb. PMID: 36252527.

3: Chan SA, Macielak RJ, Tuchscherer AM, Neff BA, Driscoll CLW, Peris-Celda M,

Van Gompel JJ, Link MJ, Carlson ML. Fluorescein-Assisted Microsurgical Resection

of Vestibular Schwannoma: A Prospective Feasibility Study. Otol Neurotol. 2022

Dec 1;43(10):1240-1244. doi: 10.1097/MAO.0000000000003718. Epub 2022 Oct 14.

PMID: 36240730.

4: Rapp CT, Amdur RJ, Bova FJ, Foote KD, Friedman WA. Repeat single-fraction

stereotactic radiosurgery for recurrent vestibular schwannoma. Rep Pract Oncol

Radiother. 2022 Sep 19;27(4):655-658. doi: 10.5603/RPOR.a2022.0073. PMID:

36196424; PMCID: PMC9521685.

5: Amit M, Xie T, Gleber-Netto FO, Hunt PJ, Mehta GU, Bell D, Silverman DA,

Yaman I, Ye Y, Burks JK, Fuller GN, Gidley PW, Nader ME, Raza SM, DeMonte F.

Distinct immune signature predicts progression of vestibular schwannoma and

unveils a possible viral etiology. J Exp Clin Cancer Res. 2022 Oct 4;41(1):292.

doi: 10.1186/s13046-022-02473-4. PMID: 36195959; PMCID: PMC9531347.

6: Dumot C, Pikis S, Mantziaris G, Xu Z, Anand RK, Nabeel AM, Sheehan D, Sheehan K, Reda WA, Tawadros SR, Abdel Karim K, El-Shehaby AMN, Emad Eldin RM, Peker S, Samanci Y, Kaisman-Elbaz T, Speckter H, Hernández W, Isidor J, Tripathi M, Madan R, Zacharia BE, Daggubati LC, Moreno NM, Álvarez RM, Langlois AM, Mathieu D, Deibert CP, Sudhakar VR, Cifarelli CP, Icaza DA, Cifarelli DT, Wei Z, Niranjan A, Barnett GH, Lunsford LD, Bowden GN, Sheehan JP. Stereotactic radiosurgery for Koos grade IV vestibular schwannoma in young patients: a multi-institutional study. J Neurooncol. 2022 Oct;160(1):201-208. doi: 10.1007/s11060-022-04134-0.Epub 2022 Sep 27. PMID: 36166113.

7: Marinelli JP, Killeen DE, Schnurman Z, Nassiri AM, Hunter JB, Lees KA, Lohse

CM, Roland TJ Jr, Golfinos JG, Kondziolka D, Link MJ, Carlson ML. Spontaneous

Volumetric Tumor Regression During Wait-and-Scan Management of 952 Sporadic

Vestibular Schwannomas. Otol Neurotol. 2022 Oct 1;43(9):e1034-e1038. doi:

10.1097/MAO.0000000000003651. Epub 2022 Aug 20. PMID: 36001695.

8: Tan NC, Macfarlane R, Donnelly N, Mannion R, Tysome JR, Jefferies S, Bance M,

Axon PR. A 2 and 5-Year Longitudinal Analysis of 671 Consecutive Patients

Diagnosed with Unilateral Vestibular Schwannoma. Otol Neurotol. 2022 Jul

1;43(6):702-708. doi: 10.1097/MAO.0000000000003536. PMID: 35709433.

Round 2

Reviewer 1 Report

Thank you for revising the manuscript according to my suggestions!

Reviewer 2 Report

This paper has been adequately redone. It remains somewhat sloppy in presentation and the authors have not been thorough in their proofreading. But the paper is informative and well enough presented now. Individual authors seem to have written their specific section but it also seems that the lead author has not gone over the manuscript as whole. I recommend accepting with corrections below. The valuable information presented outweighs remaining organizational sloppiness and organizational weaknesses. 

Required changes/corrections below:

Line 402, “In contrast, Schwann cells were represented in an inverse proportion, and

other cell types showed variable representation between the patients.” Cryptic. SC inverse proportion to what ? Rephrase.

Line “CAMP-dependent type I regulatory subunit alpha (PRKAR1A) genes…” should read

“cAMP-dependent type I regulatory subunit alpha (PRKAR1A) genes…” Note that in modern medical English use, even when a term like this begins a sentence it still retains the lower case starting letter.

Line 77, re. “bilateral tumor lesion occurs…” better would be “…bilateral VS are frequently seen”.

Lines re. 292, 331, 402… SC was already defined on line 121.

Line 442, and again on 592 you define TNF. You have already defined TNF on line 378.  Dont repeat long form.

Line 620, bevicizumab is a generic term. It starts with lower case b. 

Line 630 is unintelligible. What does “ In this regard, protein kinase inhibitors used in the targeted treatment of various neoplasias may serve as a "substitute" for its function.” Maybe you mean a substitute for merlin ? If so say so directly.

Lines 639 to 649 is intolerable. Have you switched to speaking of tumors or are you still speaking of VS.

_________________________________________________________________

Several thoughts below are optional thoughts and optional minor improvements. None of

these need be implemented. 

Fig 1 is excellent. It adds greatly. If the authors wanted they could add to the schematic the position of internal auditory meatus in temporal bone to better orient readers.

Fig 2. Is wonderful. A minor matter, “pointing to macrophages nuclei.” should read either “pointing to macrophage nuclei” or “pointing to macrophages’ nuclei.” 

Lines 205-210, Are the authors saying VS represents an aspect of  “failure of nerve regeneration” or an over exuberant repair process ? The point they make of VS occurrence at points of friction against bone, ligaments, etc is excellent.

Line 93, Potential for VS’s paracrine damage to CNVIII is an important point. Maybe repeat this in Abstract ?

Line 111, again, once you have defined vestibular schwannoma as VS, dont revert to long form.]

but line 114 is correct as it stands. In the context here, spelling the search term out [“vestibular schwannoma”] is correct.

Line 117, why stop at year 2000 ? I have experience having written over 100 peer-reviewed articles with similar data review as in this paper and I can assure you that once in a while you will find a great gem in the older literature. Go back as far as you can in PubMed. In 2 different papers I have found such gems from the 1800s ! Also several further reports from the 1070s have been useful over the years. The yield going back as far as you can in PubMed is low but

occasionally gives additional useful information. Going back as far as possible is tedious but to sustain you in that effort remember you are working for sick people.

Is line 138 correct ? VEGF participates in stimulating cells that will become new vessels. “Attract new vessels” sounds odd. 

Line 230, why is “eventual” in bold ?

Line 227, It is usual to not have comma in “phosphatase and tensin homolog”.

Lines 359, 360, “We contributed to a better understanding of the function of cancer-associated fibroblasts, supporting the theory of cancer stem cells [55–59]. “ is cryptic. Cant stem cells function without fibroblasts ?

Line 379,  re M1, M2, I read the research data to indicate both can or are potentially growth facilitating during cancer progression. I understand that you have stated the current mainstream opinion though.

Dubey S, Ghosh S, Goswami D, Ghatak D, De R. Immunometabolic attributes and mitochondria-associated signaling of Tumor-Associated Macrophages in tumor microenvironment modulate cancer progression. Biochem Pharmacol. 2022 Dec 5:115369. doi: 10.1016/j.bcp.2022.115369. 

Lines 383, 384 are cryptic. “Staining of Arginase-1, a marker of the M2 macrophage subtype, suggested strong immunoreactivity in both model groups.” What does immunoreactivity refer to ? immunohistochemical demonstration of arginase ? 

Or what ?

No need to change your text but FTI re line 648, I disagree with the proposed ASA dose. It is way too low to deeply suppress PGE synthesis. What Reckamp et al say about celecoxib holds for ASA.

Reckamp KL, Krysan K, Morrow JD, Milne GL, Newman RA, Tucker C, Elashoff RM, Dubinett SM, Figlin RA. A phase I trial to determine the optimal biological dose of celecoxib when combined with erlotinib in advanced non-small cell lung cancer. Clin Cancer Res. 2006 Jun 1;12(11 Pt 1):3381-8. doi: 10.1158/1078-0432.CCR-06-0112. 
